# An Information System Supporting Insurance Use Cases by Automated Anomaly Detection

**Thoralf Reis \*** , **Alexander Kreibich** , **Sebastian Bruchhaus** , **Thomas Krause** , **Florian Freund** , **Marco X. Bornschlegl** and **Matthias L. Hemmje**

Faculty of Mathematics and Computer Science, University of Hagen, Universitätsstrasse 1,
D-58097 Hagen, Germany
**\*** Correspondence: thoralf.reis@fernuni-hagen.de

**Abstract:** The increasing availability of vast quantities of data from various sources significantly impacts the insurance industry, although this industry has always been data driven. It accelerates manual processes and enables new products or business models. On the other hand, it also burdens insurance analysts and other users that need to cope with this development parallel to other global changes. A novel information system (IS) for artificial intelligence (AI)-supported big data analysis, introduced within this paper, shall help to overcome user overload and to empower human data analysts in the insurance industry. The IS research's focus lies neither in novel algorithms nor datasets but in concepts that combine AI and big data analysis for synergies, such as usability enhancements. For this purpose, this paper systematically designs and implements an AI2VIS4BigData reference model to help information systems conform to automatically detect anomalies and increase its users' confidence and efficiency. Practical relevance is assured by an interview with an insurance analyst to verify the demand for the developed system and derive all requirements from two insurance industry user stories. A core contribution is the introduction of the IS. Another significant contribution is an extension of the AI2VIS4BigData service-based architecture and user interface (UI) concept on AI and machine learning (ML)-based user empowerment and data transformation. The implemented prototype was applied to synthetic data to enable the evaluation of the system. The quantitative and qualitative evaluations confirm the system's usability and applicability to the insurance domain yet reveal the need for improvements toward bigger quantities of data and further evaluations with a more extensive user group.

**Keywords:** information systems; insurance analytics; big data applications; artificial intelligence; data visualization

## 1. Introduction and Motivation

Increasing data availability, improved tooling, and computational infrastructure drive progress in big data analysis. This progress enables process automation, improved product design, and increased efficiency in many application domains. The insurance industry was always data driven [1]. Thus, this industry benefits from this development. The available data in the insurance industry comprises, for instance, data from fitness trackers, meteorological data, and insurance contract data by the insurance companies themselves [2]. The internal data alone already falls into the big data category of volume [3]. Boobier estimated that insurance contracts in Great Britain alone create approximately 900 million pages per year, equating to one-fifth of the British Library [2]. Insurers apply big data analysis for use cases such as creating more personalized products [1,4], or customer acquisition [1,2,4]. Other use cases are insurance fraud detection [1,2,4], effective underwriting [2], improving processes like risk management [5], or creating other data-driven services [5].

The insurance industry benefits from the huge quantity of data by training precise models for various use cases. Artificial intelligence (AI) and machine learning (ML) support

this development in the insurance industry as it "enables many actions to be taken without explicit human instruction" [4]. On the other hand, AI and ML introduce challenges by insufficient transparency [4] and new skill requirements for insurance company employees [2]. The latter is challenging, as qualified personnel in big data analysis, AI, and ML are rarely available [6]. Thus, insurance companies need to qualify existing employees and recruit new ones simultaneously.

In addition to the need for qualification, recent global developments like climate change, inflation, recession, and the 2022 energy crisis impact insurance companies and their analysts. Climate change leads to higher damage claims or the need to update products due to more frequent severe weather conditions, such as floods [2]. A projection backs this that annual flood-related insurance claims in continental Europe will rise from €4.5 billion between 2010 and 2020 to €23 billion in 2050 [2]. Inflation and recession impact the solvency of private customers or companies, leading to a default of payment or increased insurance fraud attempts. Skyrocketing energy prices in 2022 will intensify this development. These events caused increased effort and significant changes in the insurance industry. Analyzing available data and being able to identify abnormal events thus becomes more important than ever.

The authors of this paper introduced the AI2VIS4BigData Reference Model for information systems (IS) that applies AI and ML for visual big data analysis [7]. In 2022, a service-based architecture following this reference model was introduced [8], yet never applied for a real-world use case. This paper addresses this shortcoming by implementing a practical use case and extending the reference architecture accordingly.

This work aims to implement an IS that supports experts and end users within the insurance industry to automatically detect anomalies and empower them by clearly explained and communicated results. Therefore, it aims to address the problems of overconstrained insurance analysts through increasing workload and complexity. To systematically reach this objective, the user-centered design (UCD) approach [9] is applied. This approach includes a professional insurance analyst in every step, from design to implementation to evaluation. The intended results comprise insurance industry user stories, use cases, requirements, an IS concept and implementation that fulfills these requirements, and qualitative and quantitative evaluations of the prototype implementation. The paper's objective is to contribute to state of the art in IS research for IS that apply AI and ML for big data analysis. Thus, it is strongly connected to these research domains although it aims to practically validate and extend theoretical concepts in IS research rather than defining or applying AI and ML algorithms or assessing a specific big data dataset.

The following sections consist of an overview of the state of the art with regard to data analytics for the insurance industry, anomaly detection, and the AI2VIS4BigData reference model and architecture (Section 2). Section 3 covers the conceptual model, and Section 4 discusses the proof-of-concept implementation. Its qualitative and quantitative evaluation is described in Section 5. The paper concludes by summarizing its contributions and defining objectives for future research (Section 6).

## 2. Related Work

Developing a system aiming to support individual users in insurance companies requires an understanding of the insurance industry and the importance of data analytics. As the target system shall detect anomalies and its implementation shall comply with the AI2VIS4BigData reference model, this section also reviews related work, including recent and latest papers on these topics.

### 2.1. Data Analytics for Insurance Companies

The insurance industry has existed for several hundred years, with the first so-called "insurance office" being mentioned in London in 1666 [2]. Insurance companies offer a wide range of products allowing customers to ensure indemnity in case of damaging events. "The principles of insurance [. . .] remain substantially unchanged" [2] until today.

The insurance premium's height depends on the indemnity's height, the likelihood of the occurrence of the damaging events, the impact of these damaging events, and competition within the insurance industry. Precise probability assessments by analyzing the data are key to successful products. Data analytics is thus nothing new to the insurance industry.

Boobier [2] divides the insurance industry into the five areas shown in Figure 1. Intermediaries are "third parties [that] help [insurance companies to] discharge their obligations, or optimize their operations" [2]. Examples include agents selling insurance products or contracted repairers taking care of damaged goods [2]. Specialty insurances are focused on specific domains that require a high degree of specialization and domain knowledge like, e.g., "terrorism, marine, [and] fine art" [2]. Reinsurance is "purchased by an insurance company" [2] and protects against high indemnity liabilities. Life, health, and general insurances are well known to private insurance customers and do not require a further introduction.

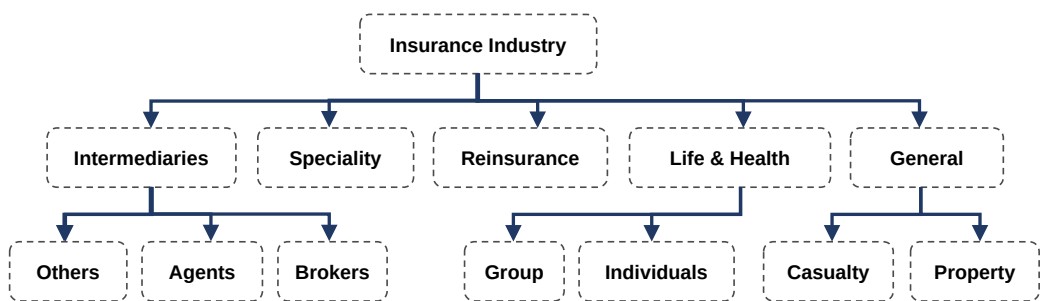

**Figure 1.** Taxonomy of the insurance industry according to Boobier [2].

Data analytics can be applied to all areas of the insurance industry outlined in Figure 1. Potential use cases comprise automated coverage advice [5] (intermediaries), maritime pirate activity monitoring [2] (specialty), or improved solvency management [2] (reinsurance). Further exemplary use cases are automated medical bill processing [5] (life and health) and estimation of probability for auto insurance claim occurrence [10] (casualty and property). For the latter example use case, the ML algorithms random forest (RF), K-Neareast Neighbors (KNN), and naive Bayes (NB) were applied to real-world auto insurance data from a Brazilian insurance company [10]. Universal insurance use cases such as detailed advertisement via improved personalization, fraud detection [4], effective underwriting [2], and improved risk management [5] can help to create value and improve efficiency in all areas of the insurance industry. Figure 2 visualizes the practical information flow for selected example insurance use cases based on big data analysis.

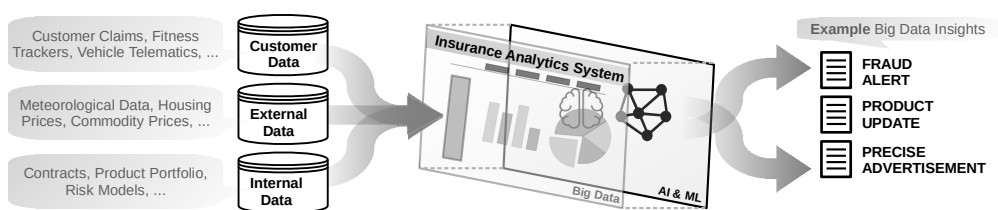

**Figure 2.** Information flow, for example, insurance use cases of big data analysis.

Figure 2 shows the different potential types of input data (customer data, external data, internal insurance data) on the left side and their transformation into exemplary big data insights on the right side. It reveals customer data-related boosters for big data analysis in insurance companies: devices that record data from the insurance customers' daily lives. Fitness trackers and smartwatches record massive quantities of data about their users' health and behavior [4], such as "walking and running" [2]. Special applications installed via software or external onboard diagnostics (OBD) devices in vehicles [2] can record the car users' driving style and car usage (e.g., trip duration, speed, and distance) [4]. This data enables improved insurance products, such as "pay-as-you-drive insurance" [4] or reward

risk-reducing behavior [4]. Insurance customers can be incentivized to provide the data to the insurance companies in exchange for discounts or reward benefits [4]. An important factor to be considered for insurers is that the willingness to share information (WSI) differs for different information categories [5]. A study by Pugnetti and Seitz [5] with 1542 Swiss insurance customers revealed that the WSI is among the highest for the vehicle information category with a WSI of 4.08 on a scale from 1 to 6 (with 6 indicating no problem with sharing information) [5]. The fitness tracker relevant information categories health monitoring (WSI of 3.39) and daily schedule information (WSI of 2.51) score lower [5]. FinTechs, InsureTechs, and their disruptive approaches in areas like, e.g., blockchain applications [2] will intensify the importance of data even further.

Increasing quantities of data raise the demand for data harmonization and standardized conceptual models. Insurers can benefit from using unified ontologies [4]. "An insurance ontology can play the role of a common, standardized vocabulary that helps communication and knowledge exchange between insurance partners" [1]. For this purpose, Koutsomitropoulos and Kalou introduced an ontology web language (OWL) ontology that represents the processes in insurance companies and enables one to "simplify analytics and deduce implicit facts" [1].

The presented information demonstrates how the already data-driven insurance industry benefits significantly from the availability of big data and suitable analytics tools. Because these technologies and trends have arrived in this industry, it remains an open question whether additional AI-based support for insurance big data analysis is necessary and beneficial.

### 2.2. Anomaly Detection

Anomalies, also known as outliers, have been subject to statistical and data analysis research for many decades [11,12]. Common definitions comprise anomalies as data instances that differ in a striking manner from other instances [11] or "observation[s] which deviat[e] so much from other observations as to arouse suspicions that a different mechanism generated it" [12]. Consequently, Chandola et al. define anomaly detection (AD) as the search for unusual patterns in data that are not in line with the "well defined notion of normal behavior" [13].

Anomalies can be divided into point, context, and collective anomalies [13]. Point anomalies differ globally from all normal instances [13]. Context anomalies deviate concerning a local contextual environment [13]. Collective anomalies appear normal as single instances yet can be exposed as abnormal within an ensemble [13]. As anomalies in time series [14] have high practical relevance, they can be named as the fourth type of anomalies, although the three other types could also cover them. The following Figure 3 shows the resulting taxonomy of big data anomaly types.

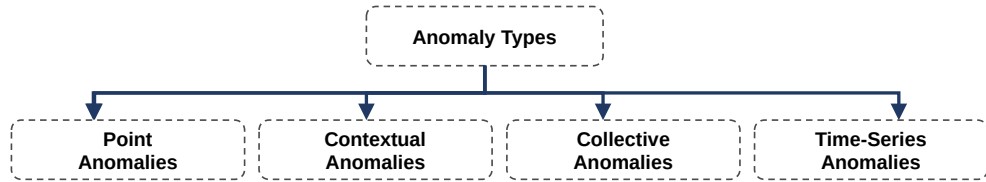

**Figure 3.** Taxonomy of big data anomaly types [13,14].

The anomalies shown in Figure 3 can be detected by applying ML-based detection strategies. These strategies are supervised, semi-supervised, or unsupervised ML [13]. Supervised AD is based on ML model training with data containing normal and anomalous data instances [13]. Because anomalies are rare, semisupervised AD is a more practical alternative [13,15]. It is based on ML model training with only normal data instances as training data [13]. Example algorithms for supervised and semisupervised AD are neural networks or support vector machines (SVM) [13]. Unsupervised AD models do not require

any labeled training data [13]. An example algorithm is KNN [13]. The three ML strategies for AD are shown in Figure 4.

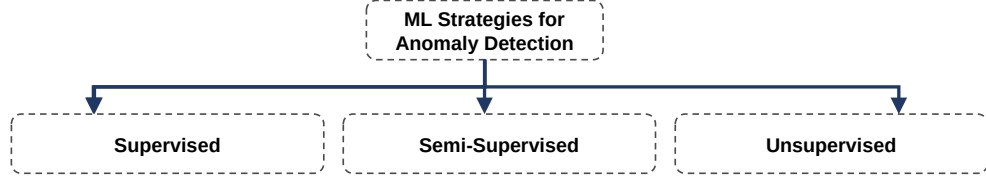

**Figure 4.** Taxonomy of ML strategies for AD [13,15].

Several application-specific algorithms exist for time-series, contextual, and collective AD [13,15–17]. A popular generic approach is to transform the problem into a point anomaly detection (PAD) and then apply a PAD algorithm [13]. PAD algorithms can follow classification-based, nearest neighbor-based, cluster-based, statistical, information theoretic-based, and spectral-based methods [15]. Classification-based methods separate the data according to their degree of normality and identify instances not belonging to a normal class as anomalies [15]. Nearest neighbor-based (or distance-based) methods assume a metric on the data and therefore declare points far from their neighbors [15]. Clustering-based methods determine instances that do not belong to a cluster or belong to smaller cluster as anomalies [15]. Statistical methods work based on the assumption that anomalies are "associated to low probability states" [15] of a data-generating process. Information theoretic-based methods determine instances or ensembles contributing disproportionately concerning their volume to the complexity as anomalies [15]. Spectral-based methods "rely on the assumption that it is possible to embed the data into a lower dimensional subspace" [15] with a more precise separation of normal data instances and anomalies. The taxonomy of AD methods is shown in Figure 5.

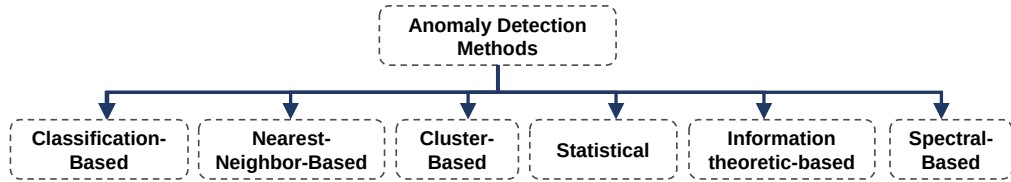

**Figure 5.** Taxonomy of AD methods [13,15].

Many publications focus on AD approaches, algorithms, and their application by data scientists or ML experts. In contrast to this big quantity of publications, the topic of IS that support end users with AD results remains an open field for future research. Research in this area is closely linked to the emergent topic of explainable AI (XAI). A reason is that IS success depends on the end users' trust in the system's results and the transparency that explanations provide is a key component of trustworthy systems.

### 2.3. AI2VIS4BigData Reference Model and Architecture

The AI2VIS4BigData reference model provides guidelines and standardizes terminologies and relationships for artifacts, components, and subcomponents of IS that aim to apply AI and ML for visual big data analysis [18]. It was introduced by the authors of this paper in 2020 [19] and is based on Bornschlegl's IVIS4BigData reference model [20] for visual big data analysis and an AI system lifecycle [18]. The reference model is shown in Figure 6.

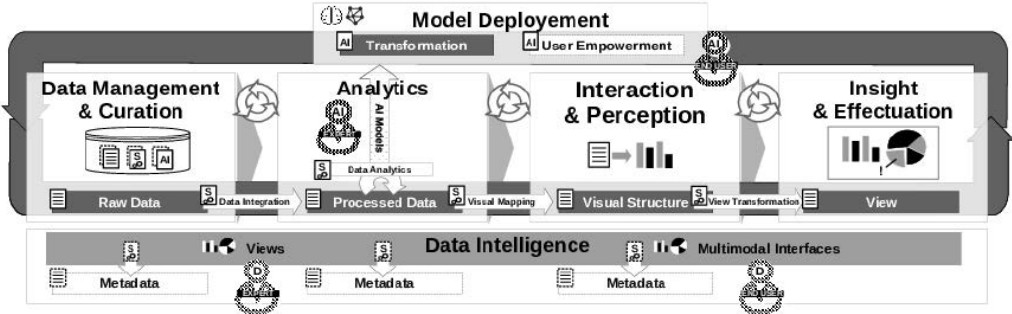

**Figure 6.** AI2VIS4BigData reference model, © 2022 IEEE reprinted, with permission, from Reis et al. [8].

The reference model consists of a pipeline with four process steps: data management and curation (1), analytics (2), interaction and perception (3), and insight and effectuation (4) [8]. The process steps are consecutively connected via the data-transformation services of data integration (1 to 2), data analytics (within 2), visual mapping (2 to 3), and view transformation (3 to 4) [8]. In addition, user empowerment services create metadata for each process step that improves understanding for the different user stereotypes involved in the process [8]. All process steps are connected via a feedback loop which emphasizes the iterative nature of data exploration and the underlying socialization, externalization, combination, and internalization (SECI) cycle for information and knowledge acquisition [21]. A model deployment layer above the pipeline can deploy AI and ML models to support the user for data transformation, and user empowerment [8]. User empowerment for the reference model follows Fisher and Nakakoji's multifaceted architecture of revealing data insights to the systems' end users [21].

The concept of user empowerment is also embedded in the reference model's service-oriented architecture (SOA) [21]. It is vertically structured following the four processing steps and enables data transformation, and user empowerment services within each vertical pillar [21]. In 2022, the SOA was implemented as a reference and applied for a health informatics use case [8]. The reference model still needed to be applied to use cases in the insurance industry. Furthermore, the reference architecture and user interface (UI) design lack a practical application of ML for empowering the systems' users.

*2.4. Discussion and Remaining Challenges*

Reviewing the state of the art revealed three remaining challenges (RC) in three areas of data analytics for insurance companies, anomaly detection, and the AI2VIS4BigData reference model:

- RC1. It is unclear whether the already data-driven insurance industry requires support for analyzing big data.
- RC2. There exists no IS that supports end users in detecting anomalies.
- RC3. The state of the art lacks a AI2VIS4BigData reference implementation for ML-based user empowerment.

The remainder of this paper aims to address these three challenges by analyzing the demand for such a system, by creating a concept for such a system, and by extending the AI2VIS4BigData reference model. The result of these concepts and its implementations are presented in the following sections.

## 3. Conceptual Modeling

The UCD approach aims at optimizing a system's usability and purpose by including the system's users in the design process [9]. The UCD consists of four steps: use context definition, derivation of requirements for the system, concept design, and evaluation (of the concept and implementation) [9]. The system's target users can be included in every

step to get "better user experience so that the users will be more comfortable using the product" [9]. Users can be included in the UCD phases directly (e.g., as designers or evaluators) or indirectly by analyzing their needs and characteristics.

Applying UCD for conceptual modeling of an IS that automates anomaly detection in the insurance industry requires the inclusion of the target users of this system. Thus, this paper involves one insurance specialist with a strong data analysis background. This expert participated in all steps from design to evaluation and answered questions in an expert interview. Section three is structured following the three derived RC:

- For RC1, a pre-study addresses an unclear demand for such a system in the insurance industry in Section 3.1;
- For RC2, Sections 3.2 and 3.3 address the design of an IS; and
- For RC3, Section 3.4 addresses the AI2VIS4BigData reference implementation.

*3.1. Insurance Analyst Expert Interview as Pre-Study*

In this research, a qualitative interview method of a problem-centered interview (PCI) [22] has been chosen for assessing the need of AI-based support for user sterotypes in the insurance industry. A PCI consists of open-ended questions and aims at establishing a dialogue between the interviewer and the interviewee [22]. Within this exchange, the "interviewer moves on to general and specific explorations as well as ad hoc questions" [22]. The PCI is applied with an insurance expert as the interviewee aiming to reveal insights regarding the demand of support for data analyses in the insurance industry. The interview was conducted on the 13th of October in 2022.

What is your educational background?

Insurance Analyst (I.A.): After the achievement of the university degree of mathematics, I started my career as specialist for calculation in non-life insurance. Simultaneously I widened my academic background by successfully completing the actuarial studies of indemnity insurance. In addition to that I completed further academic studies of buniness economics.

How many years of experience in the insurance industry do you have?

I.A.: Since over 15 years I work as actuary in non-life insurance most of that time especially in motor insurance.

What is the job title of your current occupation?

I.A.: The formal title of my current job position is a specialist for calculation and pricing of non-life insurance products.

What activities does that include?

I.A.: The main topics of my job description are model development, profit analysis of products and customer segments and economic forecasts of returns on invest. In addition to that there are corporate and communal customer prizing just as business simulation modelling to mention.

What changes did you monitor in your work environment?

I.A.: In modern insurance industry, the models build and managed get more and more complex to stay competitive. In any company, the balance of its assets and liabilities to the stakeholders has to be driven by experienced operators resulting in the well known and highly complex asset liability management (ALM) problem.

What is the impact of these changes?

I.A.: Many of these operational products have to be updated instantly as they are based on continuous data streams like for example telemetry data in the case of car insurance or rainfall data in the agricultural crop insurance. In any company also the balance of its assets and liabilities to the stake holders has to be managed by experienced operators resulting in a highly complex and substantial problem. All these topics surely require intelligent algorithms.

Does this need for intelligent algorithms also include support for the insurance analysts?

I.A.: Yes, assurance domain experts have to be supported by AI-based systems as modern companies tend to operate standard customer business cases automatically leading to economy of scale and efficient batch processing.

What further use cases for this AI-based support do you see?

I.A.: An insurer has to estimate optimally future cash flows under uncertainty of incoming insured events to ensure enduring solvency. Useful strategic management reports and pending reasonable decisions of markets or products can only be based on the results of complex high-dimensional analysis respecting all value driving features. All of these use cases demand AI-based exploration systems.

In the expert's opinion, the demand of AI-based user support in insurance industry with its increasing analytics complexity can be confirmed.

### 3.2. Use Context Empowering Insurance Analysts

Defining a realistic use context, consisting of user stories and use cases, is a core requirement for a successful UCD process. This paper thus starts its design phase with a practical twofold user story from the insurance industry. In the second step, the user story serves as a base to derive practical use cases for the target IS.

User Story A: Aftermath of an environmental catastrophe for property insurance. The analysis of insurance claims is essential for a property insurer operating in a region with a high risk of flooding. One of the most important tasks of a cost accountant is to identify objects within his portfolio that are particularly at risk. These insights are used to adjust the objects' insurance premiums after an actual flooding event. There needs to be more than just the height of the insurance claim; the claim history and the current insurance premium are also relevant. Therefore, the cost accountant needs to calculate the adjusted risk factor per object and compare it with the previous value. The target is to identify anomalously high-risk adjustments to initiate the calculation process for these objects. Due to the large number of insurance claims after such a flooding event in a high-risk region, this takes time and particular expertise to identify the most relevant objects. Textual explanations and visual highlightings point the cost accountant first to the most important data features and ensure that the contracts are adjusted before the next flood event occurs. In case of an experience-based suspicion, the cost accountant actively asks the system to determine an anomaly score for a specific object.

The same insurance company decides to design a new insurance product for environmental risks due to the immense success of their precisely priced flood insurance. The impact on the target IS is described in the second user story.

User Story B: Creation of a new product with AD preparation. An insurance product manager decides that insurance for farmers against crop shortfall caused by hail is an option for a potential new product. The product manager hired an AI and ML specialist to set up automated anomaly detection for insurance claims, as this increased the profitability of flood insurance products. The AI and ML specialist adds a new AD model to the system, consisting of a certain preprocessing, a suitable AD algorithm, and parameter values. The insurance company invites the AI and ML specialist to save time by editing an existing AD model and refactoring its application logic to the new domain. The insurance company emphasized that everything must be ready before the upcoming product launch.

The two user stories contain one user who actively interacts with the system: The AI and ML expert and the cost accountant. Both use the system for different purposes and exchange other information with it. Figure 7 summarizes these activities within a unified modeling language (UML) use case diagram.

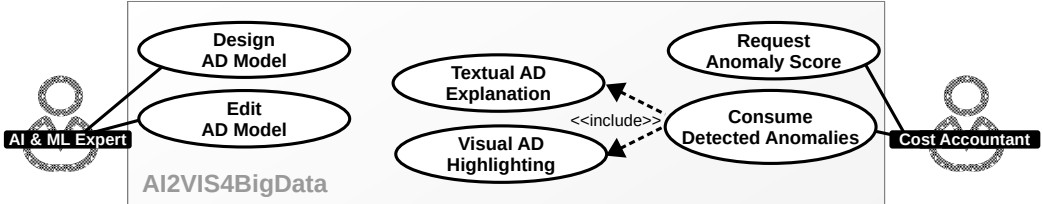

**Figure 7.** Anomaly detection use cases in an insurance company following the user stories.

Figure 7 contains six different use cases and two user stereotypes. Following the UCD process, the use case in the context of the described user stories is the foundation for deriving requirements for the target system. The derived requirements are listed below:

- (REQ1) Two independent views: The target system shall provide two independent views that reflect the different skills and demands by both user stereotypes as they use the system for disjunct use cases. These views shall focus on the AD models (AD model view for the AI & ML expert) and the actual data analytics with anomaly presentation (data analytics view for the cost accountant).
- (REQ2) Design AD models: The AD model view shall support the design and integration of an AD model. The design consists of selecting an AD algorithm, defining input data and preprocessings, parameterizing an AD algorithm and training AD algorithms (for supervised or semisupervised ML).
- (REQ3) Edit AD models: The AD model view shall support editing existing AD models and refactoring of their content.
- (REQ4) Consume detected anomalies: The data analytics view shall inform the cost accountant about anomalies in datasets that are of relevance. The detection of these anomalies shall be performed automatically without the user's active request.
- (REQ5) Textual AD explanations: Detected anomalies shall be presented to the user in a textual form. The text shall include information on the anomaly type, anomaly score, and the applied algorithm.
- (REQ6) Visual AD explanations: Detected anomalies shall be presented to the user in a visual form as a graphic. The graphic shall highlight abnormal data instances and differentiate them from normal ones.
- (REQ7) Request anomaly score: The data analytics view shall provide the function of actively requesting an anomaly score for verifying or falsifying hypotheses. The anomaly score shall be provided for selected data instances within a dataset.

The seven derived requirements are a base for further modeling and implementation activities within the following sections.

### 3.3. Modeling a User Empowering Anomaly Detection Component

The model view controller (MVC) architecture pattern divides a software system into three interacting layers [23]. It "has played an influential role" [23] since it was published in 1979 by Reenskaug [23]. The three layers of the architecture, as the name implies, are a model (data structure and persistence), a view (UI and representation), and a controller (application logic) [23]. This clear separation fits well for fulfilling the requirements from Section 3.2.

The proposed AI- and ML-based AD system contains a suitable representation of big data, corresponding data annotations, all detected anomalies, and the resources for their textual or visual communication as user empowerment in the model layer. AD services detect anomalies and generate user empowerment. These services persist in the model layer and are executed as insurance AD services for data transformation and user empowerment in the controller layer. Other elements of the controller layer are an asynchronous task queue consisting of a registry and a scheduler that manage the service execution. Two front ends communicate the AD results to the end user. One element called data investigation for requesting an anomaly score, and one element called data browser for data analytics.

The AD model designer view represents the expert interface. It enables the design and editing of AD models. Figure 8 shows the described structure of the MVC architecture.

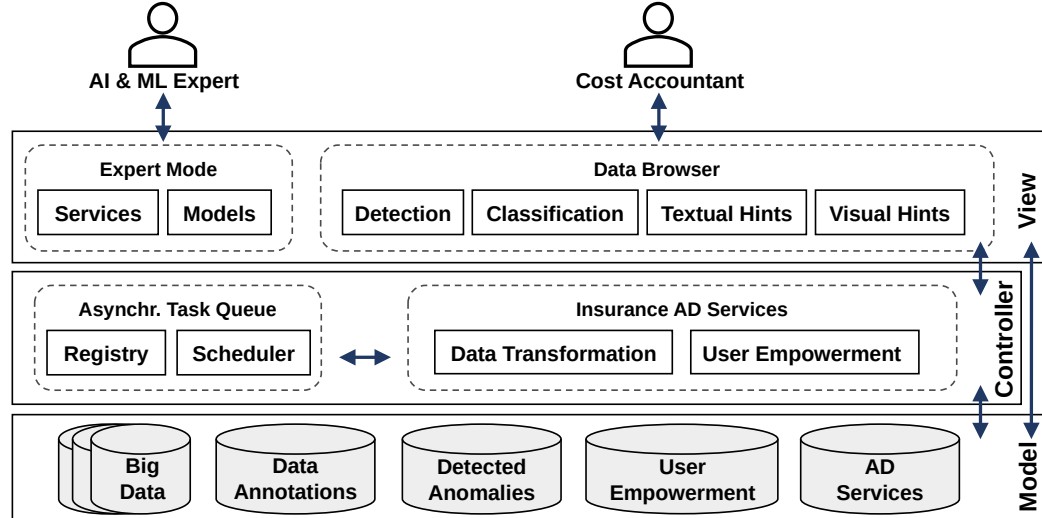

**Figure 8.** Simplified MVC architecture.

The target system shall be able to detect any anomaly type introduced in Section 2.2. This detection requires supporting different algorithms and AD methods. For managing complexity, the proposed AD system follows the favored approach of transforming AD detection into a PAP. Manual annotations of the type of transformation (contextual, collective, or time-series) by the user enable the system to still detect any anomaly. Another manual annotation for input data is determining the data type. The choice of an appropriate AD algorithm and its parameterization depends on the data characteristics, such as spatial, temporal, and high-dimensional characteristics. AD services are connected to the data annotation of analyzed datasets to use this information by selecting suitable algorithms or communicating precise results as user empowerment. Data transformation AD services detect anomalies represented in the system by a numerical score indicating its degree of abnormality. In comparison, user-empowering AD services reveal all the support information needed for content and interaction guidance via textual explanations and visual hints. Figure 9 displays a simplified version of the information model for the conceptualized system.

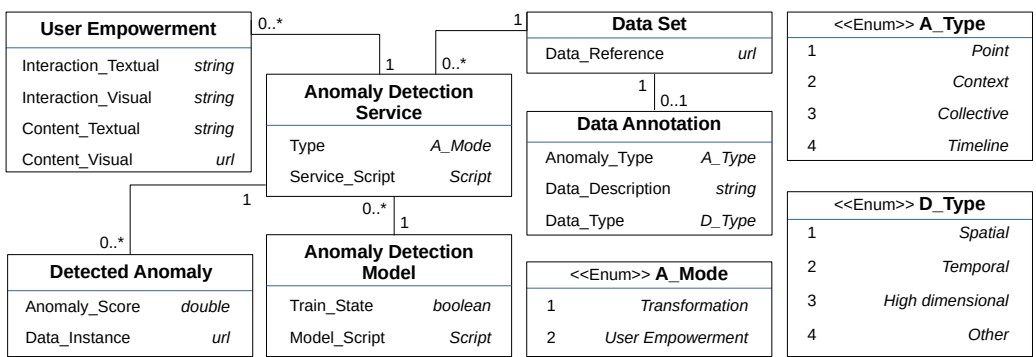

**Figure 9.** Information model for the conceptualized system.

Following the AI2VIS4BigData UI concept [24], the proposal for the system's interface contains a switch between expert and nonexpert modes to implement the two required views. It also applies the characteristic data-centric views from data integration to data exploration. The presence of anomalies is thereby actively communicated as content, and interaction guidance [24]. In addition to the AI2VIS4BigData concept, this paper proposes

to add the application-specific interactions of "detection" (manually start an AD process) and "classification" (investigate a detected or potentially abnormal data instance). It also adds whether metadata is available for a particular dataset, as user empowerment and interpretability of the detected anomalies strongly depend on it. The proposed wireframe for the UI is visualized in Figure 10.

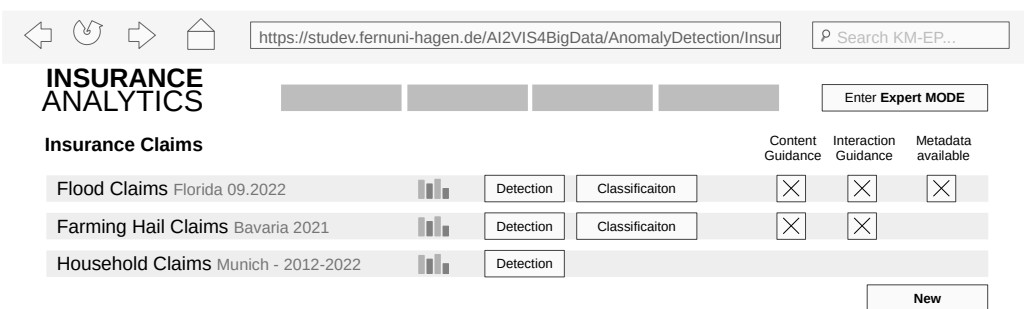

**Figure 10.** Data-centric wireframe for the cost accountant end user view.

### 3.4. Extension of AI2VIS4BigData for ML-Based User Empowerment

State-of-the-art research revealed the third and last remaining challenge of a reference implementation of the AI2VIS4BigData reference model for ML-based user empowerment. The previous section partially addressed this with the introduction of an IS for ML-based AD for user empowerment and data transformation in conformity with AI2VIS4BigData. The introduced IS differentiates within the expert view between services and models (Figure 8). This differentiation is not part of the reference model, yet a valuable and logical extension: services describe how data or metadata is created. They do not represent states such as model training or model deployment, from the life cycle of AI and ML models. Consequently, this paper proposes to extend the AI2VIS4BigData reference model with this aspect.

The concept comprises a data-centric end user view along the four processing steps from data integration to view transformation and the service-centric expert user view to manage user empowerment and transformation services. This paper proposes to add a third view: the AI and ML model-centric view for expert users. These AI and ML models shall be usable within the different services, enabling ML-based user empowerment in conformance with AI2VIS4BigData. The proposed extension of the UI concept is shown in Figure 11.

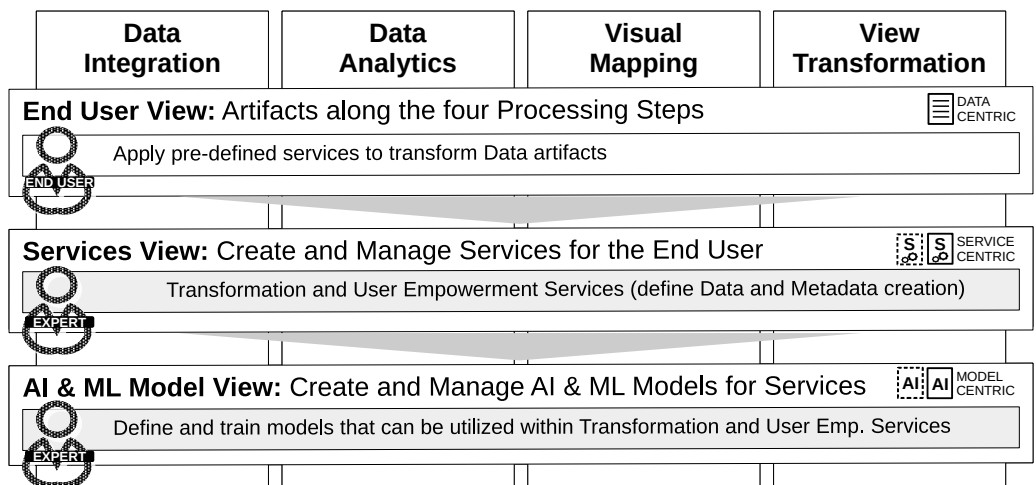

**Figure 11.** AI model extension of the UI hierarchy of AI2VIS4BigData.

Applying this extension to the insurance analytics use case investigated within this paper results in two tabs, "services" and "models", in the expert view. Via the former tab, the expert user can manage existing services and use deployed models within these services. The expert user can create new AI and ML models and manage their lifecycle from creation to deployment via the model tab. Once satisfied, the services can be released for automatic user empowerment or as usable data transformation services for end users. Figure 12 shows the wireframe of the insurance AD system incorporating this extension to AI2VIS4BigData.

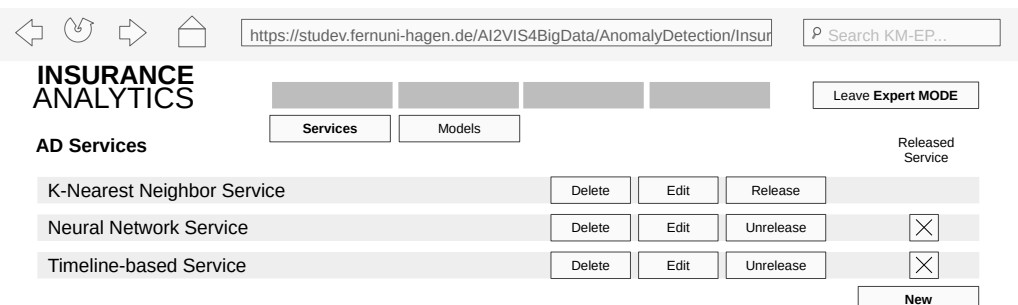

**Figure 12.** Service-centric expert user view extended with a view on AI and ML models.

## 4. Proof-of-Concept Implementation

All concepts introduced in Section 3 were implemented by the authors of this paper. This implementation was divided into three steps: the AD component, the UI, and the insurance AD use cases with the AD model application. Each of these steps is described in the following three subsections. Each of them starts with a review of existing technologies to avoid reinventing the wheel.

### 4.1. Implementation of the Anomaly Detection Component

The concepts of the AD component for insurance analytics follow the AI2VIS4BigData reference model. Consequently, reviewing existing technology starts with the reference model's existing reference implementations. With [8], the authors of this paper introduced an implementation of a service-based architecture for health informatics use cases. This architecture is the choice for the model and controller layers of this architecture with adaptions to the current application domain. This decision was motivated substantially by the following aspects. The architecture already provides generic services which can quickly implement the needed anomaly detection services and their core algorithms [8]. It integrates the workflow management system (WMS) Airflow (https://airflow.apache.org, (accessed on 7 November 2022)) as a controller core component that manages the administration and scheduling of these services [8]. It realizes a predefined data model with the Airflow-compatible structured query language (SQL) database management system (DBMS) [8] PostgreSQL (https://www.postgresql.org, (accessed on 7 November 2022)). It focuses on realizing the user empowerment process by providing components to deploy content and interaction guidance support [8].

Another core element of the controller is the AD algorithm within the data transformation and user empowerment services. The clear focus on the system on detecting anomalies eases the search for suitable AI and ML algorithms and libraries. PyOD (https://pyod.readthedocs.io, (accessed on 7 November 2022)) is a Python [25] library that provides plenty of well-known and proven algorithms for outlier detection with a particular focus on spatial data structures. PyOD contains the simple and easily understandable unsupervised KNN algorithm implemented within the prototyping activities. PyOD fulfills the requirements for a prototype in the insurance application domain and was selected for the prototype implementation.

The choice for the view of the system is the JavaScript framework Vue.js (https://vuejs.org, (accessed on 5 November 2022)). It simple, easy to learn, and enables the implementation of the wireframes presented in Section 3. Apart from standard HTML/CSS/JavaScript, Vue.js offers the techniques of "routers" and "watchers", which help implement the AI2IVIS4BigData pipeline stages. Figure 13 summarizes the selected technologies for the model, view, and controller.

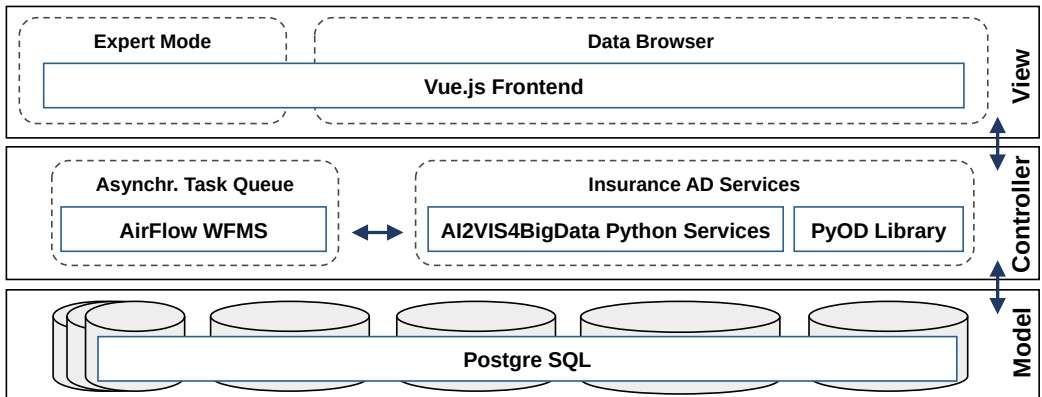

**Figure 13.** Solution architecture with applied technologies.

### 4.2. Implementation of the Graphical User Interface

As introduced in the previous subsection, the UI of the prototypical implementation was implemented as a Vue.js web application that follows the AI2VIS4BigData UI concept [24]. Consequently, the UI structure implements the four stages of the AI2IVIS4BigData development pipeline [24] with four tabs: integration, analysis, visualization, and perception. Due to the paper's scope, the authors implemented only the first two tabs in the prototype. Within the first tab, big data are imported into the AD system. Depending on the type of anomalies, this can require manual annotations (e.g., contextual transformation). Thus, the implemented data integration view enables the expert user to provide annotations for data and anomaly types. Three horizontal lines organize the UI in three areas. The topmost area allows the selection of data files and annotation of their structure as spatial and temporal data. The second area enables parameterizing optional data preprocessing by annotating the anomaly type as point, context, collective, or timeline anomaly. The last area contains all integrated data sets that can be either preprocessed or deleted. Figure 14 shows the implemented data integration view.

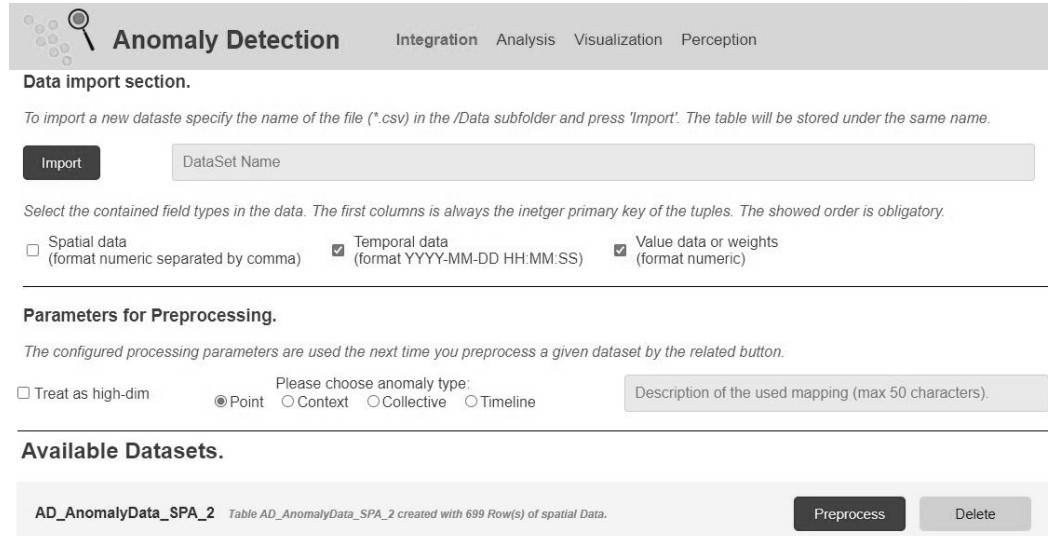

**Figure 14.** UI implementation for data integration, preprocessing, and annotation.

Integrating or preprocessing data starts the AD services to create content and interaction guidance for the end user. The second tab (analysis) communicates this user empowerment content. The user can access this information via the (green) drop-down buttons on the right side of a data entry. Displaying an SQL statement to query potentially abnormal data instances was implemented as interaction guidance. The use case of requesting an anomaly score is implemented via the two blue buttons, "detection" (perform a customized AD) and "classification" (investigate single data instances). Following the logic of the wireframes, the user can change to the expert view via a dedicated button on the top right side of the analysis page. The analysis page is shown in Figure 15.

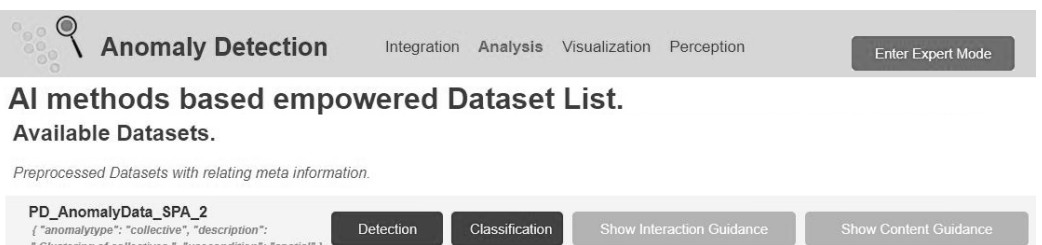

**Figure 15.** UI implementation for end user data analysis.

### 4.3. Implementation of the End User Use Case

Realistically evaluating the system, its capabilities, and benefits for end users requires the possibility of using and experiencing the system in real-world conditions. An implementation with realistic data, services that cover the planned use cases and AD models that can find apparent anomalies within the data enables such a practical evaluation. Lacking access to real-world insurance claims, the authors of this paper implemented a data generator. It nondeterministically simulates spatial data around a variable number of cluster centroids. The generator adds a random number between 200 and 250 data instances around each centroid with a random standard deviation between 1 and 25. Figure 16 shows three datasets created by the tool, each containing three cluster centroids. For better readability, the data instances of the same cluster share the same color. A rectangular hull of one standard deviation around all cluster centroids serves better orientation.

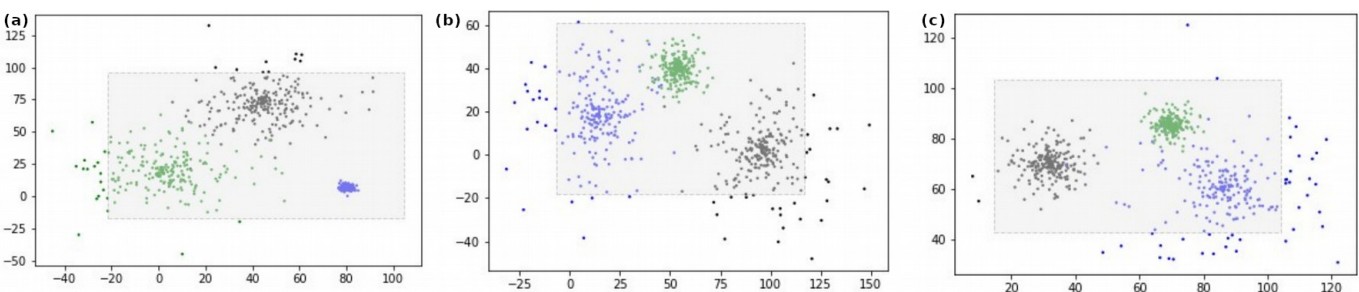

**Figure 16.** Synthetic test data generated for the insurance use case. (**a**) two widely spread and one very dense cluster; (**b**) three widely spread clusters; (**c**) one dense, one moderately spread, and one widely spread cluster.

Implementing the use cases from Figure 7 within an AI2VIS4BigData-conform system starts with a data integration transformation service. A CSV integration service was implemented as the implemented data generator stores its results in comma-separated values (CSV) files. The next step of the implemented UI (refer to Figure 14) is the preprocessing of integrated data. A preprocessing service was implemented and integrated into the system for that purpose. In addition to simulating preprocessing, it enables the annotation of the anomaly and data types. The AI and ML expert user use cases contain the creation and editing of an AD model. As the PyOD library provides a simple KNN algorithm, an

AD model using the KNN was developed, parameterized, and added to the system. The parameters for the KNN model are shown in Listing 1.

**Listing 1:** Parameters for PyOD KNN anomaly detection model.

```
n_neighbors = 5,        # five neighbors used for k neighbors queries
method = 'mean',        # average of all k neighbors as anomaly score
metric = 'Euclidean'    # euclidean distance as metric
```

The cost accountant use case of requesting an anomaly score was implemented in two steps (as Figure 15 introduced already); detection of anomalies (implemented as a transformation service); and classification of the detected anomalies (implemented as a user empowerment service). The remaining use case of textually or visually consuming detected anomalies was added to the system as a user empowerment service. Figure 17 shows the five implemented services, the AD model within their artifact category, and the AI2VIS4BigData reference model processing step.

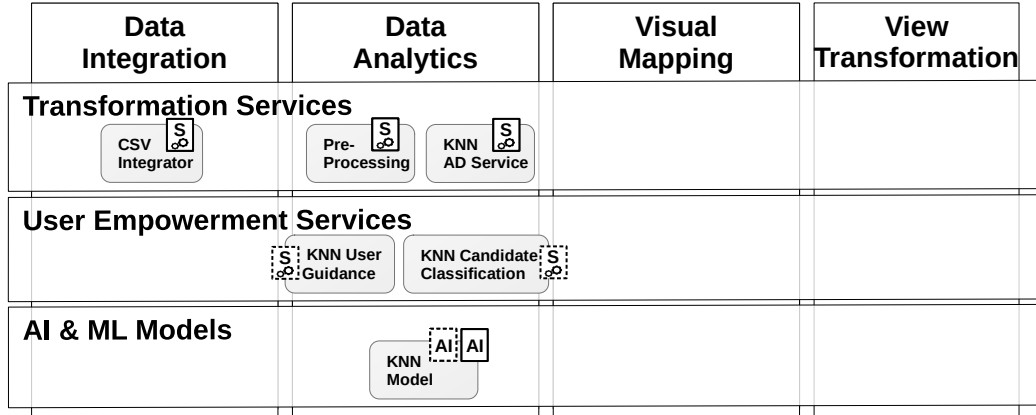

**Figure 17.** Implemented services and AD models categorized within the AI2VIS4BigData.

## 5. Evaluation

Comprehensive evaluations of the implemented prototype system and use cases consider quantitative and qualitative aspects and include real-world insurance industry users. It thereby strengthens the followed UCD approach. Consequently, this paper assesses the qualitative results of the conducted anomaly detection and quantitatively reviews the performance characteristics of the developed system. Finally, it qualitatively assesses the system's usability. The latter was evaluated by using a cognitive walkthrough (CW) [26] including such a real-world insurance industry user.

The first evaluation method (Section 5.1) assesses the detection of the applied PyOD KNN model (from Section 4.3) to ensure that the model is working correctly. It consists of an integration of graphical visualizations of detected anomalies in the system's UI. It follows a manual procedure whereby the evaluator adjusts the KNN model until the resulting anomalies are plausible. The artificially generated datasets with known anomalies serve as the baseline. The second evaluation method (Section 5.2) is a quantitative evaluation aiming to measure the impact of growing dataset sizes and, thus, the fitness for real big data applications. It compares the three created datasets (from Section 4.3) against each other and thereby creates its baseline (relative measurements). The third evaluation method (Section 5.3) is a CW that aims to validate the system's usability. It follows the protocol introduced in [27] that consists of a proband that is asked to break down a specific task into required actions. These actions are then compared to ground truth as a baseline.

### 5.1. Qualitative Evaluation of Anomaly Detection

This paper focuses on developing an IS to support the insurance industry's end users in applying AD for their daily work. It does not cover the development of new AD algorithms. Thus, assessing the performance of the used AD algorithm from the PyOD library would

not provide a benefit. Nevertheless, the system must correctly apply the algorithms. Thus, the implemented user guidance service for end users graphically displays the dataset and highlights significant anomalies in red. It creates XY graphs for all potential combinations between the data's dimensions. Significant anomalies have an anomaly score bigger than $2\sigma$ compared to the average score.

Figure 18 shows exemplarily the output of the user empowerment service for the simulated test dataset from Figure 16b. Because the data-generation algorithm and the dataset are known, the authors confirmed the correct AD results.

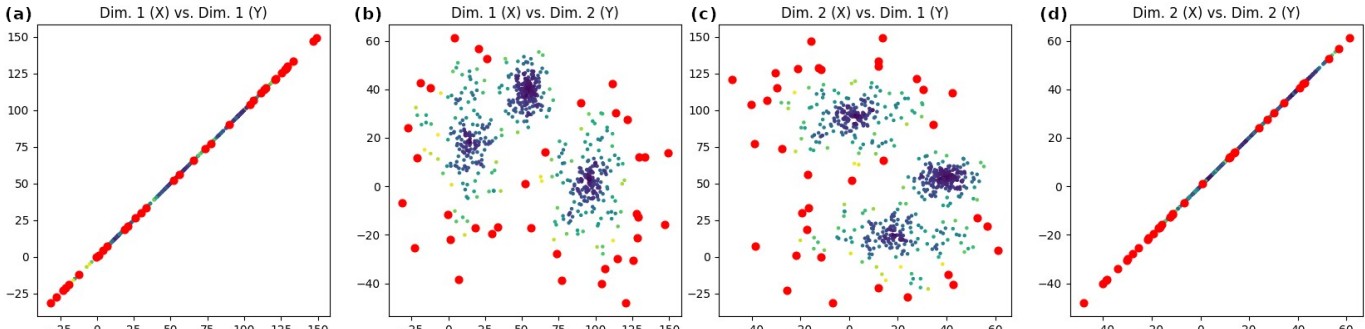

**Figure 18.** Anomalies for synthetic test data. (**a**) Dimension 1 as X, dimension 1 as Y; (**b**) dimension 1 as X, dimension 2 as Y; (**c**) dimension 2 as X, dimension 1 as Y; (**d**) dimension 2 as X, dimension 2 as Y.

The conclusions that can be drawn from this evaluation method are limited. The AD model successfully identified artificially introduced anomalies in the three test datasets. Thus, the aimed functionality of the system can be confirmed, but it enables no generalization for other data, AD models, or AD model parameters.

### 5.2. Quantitative Evaluation of the Anomaly-Detection Component

The three introduced test datasets from Section 4.3 are also used for quantitatively evaluating the system. They comprise $N_1 = 681$, $N_2 = 699$, and $N_3 = 697$ data points and thus have a similar size. For covering a regular user journey in the evaluation, this paper assesses the services for preprocessing, the KNN-based AD service, and the service for user guidance that utilizes the KNN results. The scope of the evaluation is a measurement of the respective processing time and therefore used processor share for each combination of data set and service.

Figure 19 shows the results of the processing time measurement. All measurements show similar runtimes per service for the three test datasets. This can be explained with a similar sample size. The service's different order of function explains the different average runtime durations (17.6 s, 28.6 s, and 21 s). The preprocessing does not depend on the sample size as it only adds metadata on the dataset level. It is thus $O(1)$. The computation complexity of the KNN classifier in its unoptimized form is with $O(N^2)$ [28] very high. Consequently, it required the highest computation time in the measurement. The user guidance service utilizes the anomaly score for every data instance calculated by the KNN classifier and determines the samples with a score above $2\sigma$. Thus, the standard deviation needs to be calculated. The big O notation of determining the standard deviation is $O(N)$ as it requires iterating through the dataset. All measurements were conducted via the Airflow system logs.

The processor load strongly depends on parallel tasks on the host system. The developer ran the Airflow-based proof-of-concept implementation on a personal computer; thus, the measurements are not free from impact. Table 1 displays the measured processor loads for the three test datasets and services.

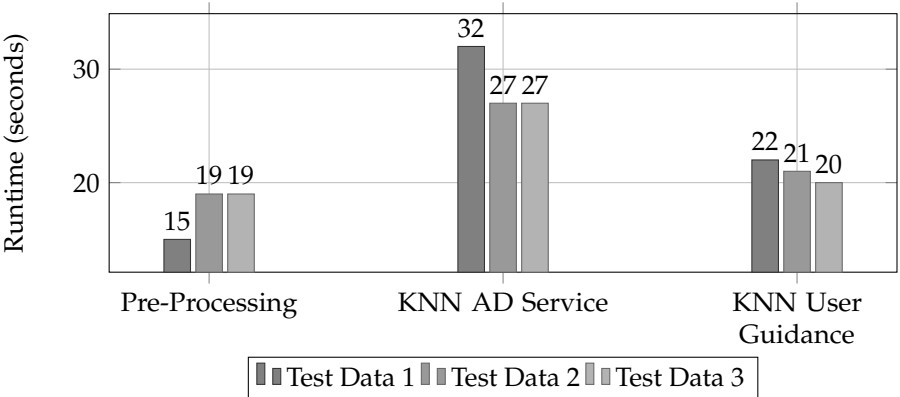

**Figure 19.** Quantitative assessment of the system implementation's runtime for three datasets.

**Table 1.** Measurement of the duration of feature extraction and inference for three test datasets.

| Test Data | | Average Processor Load [%] | | |
|---|---|---|---|---|
| Data Set | Data Points | Pre-Processing | KNN AD Service | KNN User Guidance |
| Test Data 1 | 681 | 41.51 | 45.82 | 37.73 |
| Test Data 2 | 699 | 38.77 | 45.68 | 37.73 |
| Test Data 3 | 697 | 38.52 | 42.97 | 38.53 |

The result confirms the highest computation demand for the KNN AD service. The services for preprocessing and user guidance result in similar processor consumption. In conclusion, the measurement fits the respective services' big O notation. Limiting factors for all measurements are the environmental influence on the computer, datasets with similar characteristics, and the low data volume. The processor load and high service runtimes for the KDD AD service make it challenging to be used for big data scale datasets. Another conclusion is that the services and its Airflow-based WMS are not suited for low response time application rather than execution in the background.

### 5.3. Qualitative Evaluation of the Insurance Analyst Use Case

The qualitative evaluation of the insurance industry use cases for the cost accountant end user was conducted via a CW [26]. A CW is a usability evaluation method that aims to verify information availability and system understandability from the user's perspective. It reviews how an evaluator that empathizes with a target user stereotype solves a task in comparison to the underlying system premises (ground truth). The user-centric characteristics of a CW fit well with the UCD approach followed in this paper. The initial step is the preparation of the CW with specific tasks and a breakdown into a sequence of basic operations [29]. This paper chose to hand this preparation over to an insurance domain user involved in the system development to follow the UCD paradigms. An IT manager that never used the prototype before was selected as the proband. The CW evaluation comprises two use cases: Request anomaly score and consumes detected anomalies.

The first use case of requesting an anomaly score was broken down into three tasks consisting of 19 actions. The following listing shows the three tasks:

(1)  perform a complete AD for a certain dataset;
(2)  review a data instance with an exceptional anomaly score; and
(3)  reclassify a data instance from the dataset with another AD service.

The user-empowering guidance service implemented the second use case of consuming detected anomalies. It also consists of three tasks:

(4)    preprocess data and label its data and anomaly type;
(5)    identify and interpret the graphical anomaly content guidance; and
(6)    identify and interpret the textual anomaly interaction guidance.

Table 2 displays the tasks with its ground truth number of actions, the identified actions, raised problems, and the ratio of issues per ground truth action. The table reveals that the first use case and its tasks achieved a good result: all actions were identified, and only the second task received a finding. This finding is related to the proband missing a clear system note signaling that the started AD process results are not immediately available via the UI.

**Table 2.** Cognitive walkthrough results with tasks for the insurance user stereotype.

| Task | Evaluation | | | |
|---|---|---|---|---|
| Description | Required Actions | Identified Actions | Problems | Problem Rate |
| (1) Perform a complete AD for a certain data set | 4 | 4 | 0 | 0% |
| (2) Review a data instance with an exceptional anomaly score | 7 | 7 | 1 | 14% |
| (3) Reclassify a data instance from the data set with another AD service | 8 | 8 | 0 | 0% |
| (4) Pre-process data and label its data and anomaly type | 3 | 2 | 3 | 100% |
| (5) Identify and interpret the graphical anomaly content guidance | 3 | 3 | 1 | 33% |
| (6) Identify and interpret the textual anomaly interaction guidance | 3 | 3 | 0 | 0% |
| Total | 28 | 27 | 5 | 18% |

The second use case did not perform as well. Table 2 reveals that the proband did not identify one of the nine required actions. The missed action was labeling the data type. The proband stated that it did not determine that the corresponding section of the implementation (refer to Figure 14) belongs to the preprocessing task. Another finding for the task (4) is the lack of visual feedback that the result of the user empowerment service is available. The third identified problem increases this severity: The buttons "Show Interaction Guidance" and "Show Content Guidance" (refer to Figure 15) are visible already before the results are available. The proband proposed to disable the relevant buttons while executing AD. The second task with findings comprised the obligation to visually recognize and semantically understand the generated guidance support. The proband raised the concern that the textual explanation of the number of identified instances did not match the red-labeled anomalies in the graphics. The reason for this was different thresholds that needed unification for better usability.

In conclusion, the general usability of the system can be confirmed as almost all required actions were successfully identified. However, there exists the need to improve the system toward five certain aspects to overcome identified problems by the proband. Furthermore, the CW could be extended to a larger peer group.

### 6. Summary and Outlook

This paper applied the AI2VIS4BigData reference model and its architecture to a real-world use case in the insurance industry. The research is based on recent AI2VIS4BigData publications about a service-based architecture in health informatics [8] as well as a user-empowering application in the meteorology domain [27]. It introduces a big data analysis system that supports insurance user stereotypes by automatically detecting anomalies. All steps of system development focus on a user-centered system by including a professional insurance expert.

A state-of-the-art review (Section 2) of data analytics for insurance companies, anomaly detection, and the AI2VIS4BigData reference model revealed three remaining challenges within this paper: uncertainty about the need for data analysis support in the

insurance industry, lack of an IS that supports end users detecting anomalies, and lack of an AI2VIS4BigData implementation of ML-based user empowerment. The conceptual modeling within this paper (Section 3) addresses these remaining challenges by conducting an expert interview with an insurance analyst, systematically designing the system, and extending the AI2VIS4BigData UI concept.

AI2VIS4BigData's service-based architecture is the foundation for the novel system's implementation in Section 4. This architecture is extended by combining services with AI and ML models, a new frontend, and an AD library. Finally, a prototype implementation of required services and simulated data enables the theoretically derived use cases. The evaluation in Section 5 covers both quantitative and qualitative aspects. It describes ensuring the correctness of the included AD models, measures the performance of the implemented prototype, and assesses the system's usability by applying the CW methodology.

The main contributions of this paper are an insurance expert interview, a novel IS for AI-supported data analysis in the insurance industry, and an extension of AI2VIS4BigData's UI layer on AI and ML models. The expert interview confirmed the hypothesis that analysts benefit from hints about potential data anomalies, even in the data-driven insurance industry. The information system was designed following the MVC pattern and implemented by using state-of-the-art technologies such as Apache Airflow. AI2VIS4BigData Reference Model was refined on a definition and a UI concept, how AI and ML models interconnect with data transformation and user empowerment services. All implementations were evaluated quantitatively and qualitatively. The advantages of the proposed concepts lay in their novelty. As the state-of-the-art review revealed, three significant gaps have been addressed with this work. Future use cases can be built upon these findings and the extended concepts.

The remaining challenges for future research include addressing the issues identified in the CW, applying heterogeneous and more extensive data in the system, and applying its underlying principles to other application domains. The issues identified in the CW comprise that not all elements or system states were identifiable for the proband user. Data heterogeneity and quantity limitations exist as the system was only validated by using synthetic, self-generated data. The application to other practical application domains should clarify whether the introduced user empowerment principles are helpful only for insurance domain user stereotypes or are applicable in a more generic way.

**Author Contributions:** Conceptualization, formal analysis, investigation, resources, validation, and visualization T.R. and A.K.; methodology, T.R.; software and data curation, A.K.; writing—original draft preparation, T.R. and A.K.; writing—review and editing, T.R., A.K., S.B., T.K., F.F., and M.X.B.; supervision, M.X.B. and M.L.H. All authors have read and agreed to the published version of the manuscript.

**Funding:** This research received no external funding.

**Institutional Review Board Statement:** Not applicable.

**Informed Consent Statement:** Not applicable.

**Data Availability Statement:** Not applicable.

**Conflicts of Interest:** The authors declare no conflict of interest.

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
