# Peer review of "An Information System Supporting Insurance Use Cases by Automated Anomaly Detection"

_2504-2289, doi:10.3390/bdcc7010004_

Round 1
Reviewer 1 Report
The paper is focused on presenting the design and implementation of a platform focused on supporting insurance use cases by automated anomaly detection. To accomplish this task, the authors are supported by the AI2VIS4BigData reference model, focused on applying artificial intelligence and machine learning for visual big data analysis. We think that it is relevant to point out that the authors of the current manuscript, are also the authors that formerly proposed the AI2VIS4BigData model, therefore it is interesting to discuss new case studies around such models, like the presented in the current paper.
Beyond these issues, we think that the evaluation section (Section 5) should be improved a bit, according to the following issues:
-What is the purpose of the evaluation and the used protocol?
-Is there any baselines?
-Which are the main conclusions of such section?
-Even though some of these questions have been already referred, we think that such content should be clarified, highlighting the practical implications and avoiding excessive technical language.
-Is there any antecedent to the current works, focused on the same working domain?
At this stage we suggest minor revision.
Author Response
Dear Sir or Madam,
We are very grateful for your review.
We tried to address your points as follows:
-
Extension of the introduction of chapter 5 with a new (second) paragraph that describes for each of the three evaluation methods what purpose, protocol, or baselines are followed.
-
We added a paragraph after each sub-section in chapter 5 explaining each evaluation method's main conclusions.
Besides these changes in chapter 5, the following is the complete changelog based on comments from you and other reviewers:
-
Abstract
-
Revised abstract to follow the precise pattern of 1) Background 2) Aim/Objective 3) Methodology 4) Results 5) Conclusion.
-
We added a clear statement on the research focus: neither AI algorithms nor Big Data datasets but concepts that combine both for synergies like, e.g., usability improvements.
-
-
-
1) Chapter
-
Statement on the problem, results, and contributions of the paper before presenting the paper structure (2nd last paragraph in chapter 1)
-
Adding a statement to clarify that AI/ML methods and data are not the centerpiece but that the work is nevertheless still part of AI/ML research domains (IS for applying AI and ML for visual Big Data Analysis)
-
-
-
2) State of the Art → Related Work
-
We renamed Section 2 to “Related Work” to emphasize the literature review in their study area.
-
-
3) Conceptual Model
-
No change
-
-
4) Implementation
-
Extension of the introduction to chapter 4 to emphasize the implementation of the proposed system: clarify that all components and models have been implemented and in which subsection the individual results are presented.
-
Added KNN model parameters as a listing in section 4.3
-
-
5) Evaluation
-
Extension of the introduction of chapter 5 with a new (second) paragraph that describes for each of the three evaluation methods what purpose, protocol, or baselines are followed.
-
We added a paragraph after each sub-section in chapter 5 explaining each evaluation method's main conclusions.
-
-
6) Summary and Outlook
-
Reference antecedent work that is the foundation for the presented study in the first paragraph of chapter 6.
-
Rewriting the contribution paragraph to summarize the paper's conclusions and essential findings in a way that helps other researchers.
-
Added statement of advantages of the presented method against others
-
Adding an explanation of the limitations of the presented study.
-
Thank you very much for your subsequential review and your time.
Best Regards,
Thoralf Reis, on behalf of all authors
Reviewer 2 Report
The paper proposes to develop to develop an AI-based Information System that will help insurance companies detect fraud, advertisement, etc.
Major concerns:
1) This is neither an AI/ML research paper nor a big data research paper as the name/title suggests. Authors should consider re-writing/re-structuring the paper and re-submitting to a journal that focuses on software design or customer survey analysis. Because, the only research work done here is 1) the customer survey. They propose a MVC user interface as well but there is not much justification for their particular design
2) AI/ML proposed is not innovative. Even the data used is small and more alarmingly synthetic data with no results from real world dataset. Authors might also consider re-submitting this as a review paper.
3) Figures 2-5 should be combined in 1 figure; Figure 6 is not viewable
Author Response
Dear Sir or Madam,
We are very grateful for your review. We had long discussions upon your feedback to reorientate toward another Journal due to the lack of Big Data or AI relevance. We compared it with previous contributions to BDCC from our University and got feedback from the MDPI editors that they would accept the content for this journal. We understand your opinion: The paper is not a classical use case paper but instead describes the indirect relationship between Big Data and AI from the perspective of an Information System. As the main objective of the developed Information System is the analysis of Big Data, we still argue the fit for this Journal.
As we take your critical feedback very seriously, we tried to implement several changes (starting with a more precise title) that explain the research objective of the paper better and, thus, hopefully, make it easier to accept it for BDCC.
Here is the detailed list of changes:
-
General
-
Adjusted the title to portray the actual intent of the study: the creation of an information system for Big Data Analysis for a practical application domain (rather than a specific use case).
-
Review of Figures 1,3,4,5 to see if merging improves the understandability of the paper. Conclusion: keep them different figures as they are all disjunct taxonomies (not interconnected). Combining them into one picture would make it more complex and they could not be presented within the matching text sections. We thus decided to keep them as they are. Figure 6 (the reference model) is a reprint from IEEE / MDPI publications and should be high quality in the PDF version.
-
-
Abstract
-
Revised abstract to follow the precise pattern of 1) Background 2) Aim/Objective 3) Methodology 4) Results 5) Conclusion.
-
We added a clear statement on the research focus: neither AI algorithms nor Big Data datasets but concepts that combine both for synergies like, e.g., usability improvements.
-
-
1) Chapter
-
Statement on the problem, results, and contributions of the paper before presenting the paper structure (2nd last paragraph in chapter 1)
-
Adding a statement to clarify that AI/ML methods and data are not the centerpiece but that the work is nevertheless still part of AI/ML research domains (IS for applying AI and ML for visual Big Data Analysis)
-
-
2) State of the Art → Related Work
-
We renamed Section 2 to “Related Work” to emphasize the literature review in their study area.
-
-
3) Conceptual Model
-
No change
-
-
4) Implementation
-
Extension of the introduction to chapter 4 to emphasize the implementation of the proposed system: clarify that all components and models have been implemented and in which subsection the individual results are presented.
-
Added KNN model parameters as a listing in section 4.3
-
-
5) Evaluation
-
Extension of the introduction of chapter 5 with a new (second) paragraph that describes for each of the three evaluation methods what purpose, protocol, or baselines are followed.
-
We added a paragraph after each sub-section in chapter 5 explaining each evaluation method's main conclusions.
-
-
6) Summary and Outlook
-
Reference antecedent work that is the foundation for the presented study in the first paragraph of chapter 6.
-
Rewriting the contribution paragraph to summarize the paper's conclusions and essential findings in a way that helps other researchers.
-
Added statement of advantages of the presented method against others
-
Adding an explanation of the limitations of the presented study.
-
Thank you very much for your subsequential review and your time.
Best Regards,
Thoralf Reis, on behalf of all authors
Reviewer 3 Report
Title: Supporting Insurance Use Cases by automated Anomaly Detection.
The authors have presented supporting insurance use cases by automated anomaly detection. The paper is well written, and in a way, the practical analysis used supports the presented models due to my own observation, and the paper is also relevant to the journal. However, the author has to look into the following concerns:
1. The title needs to be rewritten in a proper way, the present title did not really portray the real intent of the study.
2. Authors are advised to be precise in the abstract, and structure your abstract as follows- 1) Background 2) Aim/Objective 3) Methodology 4) Results 5) Conclusion. Write 2-4 lines for each and merge everything in one paragraph (200-300 Words) without any subheading.
3. The introduction section is very short and did not present the problem the paper wants to solve clearly, the contributions are not well stated, therefore, the paper is very difficult to follow. Also, the motivation and contribution should be stated more clearly at the end of the introduction immediately before the structure of the paper.
4. The authors are advising creating a section called related work to give an updated and complete literature review in their area of study. Recent latest papers which studied similar effects problems can be discussed to help the readers.
5. The proposed system is not being implemented, if yes kindly explain the implementation of the proposed system. The model should be simulated to really show how it works.
6. It will be worth mentioning if the author can state the advantages of the chosen methods against others.
7. The author seems to disregard or neglect some important findings in the results that have been achieved in the paper. So elaborate and explain the results in more detail.
8. It would be interesting that the author explains the limitations of the present study to help other authors for future studies. Mention the future scope of your present works.
9. The conclusion should be rewritten; the author should be able to elaborate on their work in the conclusion part so as to help the readers understand the work. The results from the study should also be explained in the conclusion part. The future directions for this study will help readers who want to work in this area.
I really appreciate the style of presentation of this paper, but the author needs to incorporate the above-mentioned points for a better and possible publication with the journal. I, therefore, recommend a minor revision.
Author Response
Dear Sir or Madam,
We are very grateful for your review.
We tried to address your points as follows:
- We have adjusted the title to portray the actual intent of the study: the creation of an information system for Big Data Analysis for a practical application domain (rather than a specific use case).
- Revised abstract to follow the precise pattern of 1) Background 2) Aim/Objective 3) Methodology 4) Results 5) Conclusion.
- Extension of Chapter 1 with statement on the problem, results, and contributions of the paper before presenting the paper structure (2nd last paragraph in chapter 1 and by adding a statement to clarify that AI/ML methods and data are not the centerpiece but that the work is nevertheless still part of AI/ML research domains (IS for applying AI and ML for visual Big Data Analysis)
- We renamed Section 2 to “Related Work” to emphasize the literature review in their study area.
- Extension of the introduction to chapter 4 to emphasize the implementation of the proposed system: clarify that all components and models have been implemented and in which subsection the individual results are presented. Added KNN model parameters as a listing in section 4.3
- Added statement of advantages of the presented method against others in Chapter 6
- Extension of the introduction of chapter 5 with a new (second) paragraph that describes for each of the three evaluation methods what purpose, protocol, or baselines are followed. We added a paragraph after each sub-section in chapter 5 explaining each evaluation method's main conclusions.
- Extension of Chapter 6 with an explanation of the limitations of the presented study. Added statement of advantages of the presented method against others
- Rewriting the contribution paragraph to summarize the paper's conclusions and important findings in a way that helps other researchers.
Thank you very much for your kind words with your first review, the supporting comments, the subsequential review, and the time.
Best Regards,
Thoralf Reis, on behalf of all authors
Round 2
Reviewer 2 Report
The authors have significantly addressed reviewers comments.